# Comparative Analysis of the Minimum Number of Replication Origins in Trypanosomatids and Yeasts

**DOI:** 10.3390/genes11050523

**Published:** 2020-05-08

**Authors:** Marcelo S. da Silva, Marcela O. Vitarelli, Bruno F. Souza, Maria Carolina Elias

**Affiliations:** Laboratório de Ciclo Celular, Center of Toxins, Immune Response and Cell Signaling (CeTICS), Instituto Butantan, São Paulo 05503-900, Brazil; marcela.vitarelli@butantan.gov.br (M.O.V.); fsouza.bruno@gmail.com (B.F.S.)

**Keywords:** trypanosomatids, yeasts, trypanosomatids genome, cell cycle phases, S-phase duration, DNA replication, replication origins

## Abstract

Single-celled eukaryote genomes predominantly replicate through multiple origins. Although origin usage during the S-phase has been elucidated in some of these organisms, few studies have comparatively approached this dynamic. Here, we developed a user-friendly website able to calculate the length of the cell cycle phases for any organism. Next, using a formula developed by our group, we showed a comparative analysis among the minimum number of replication origins (MO) required to duplicate an entire chromosome within the S-phase duration in trypanosomatids (*Trypanosoma cruzi*, *Leishmania major*, and *Trypanosoma brucei*) and yeasts (*Saccharomyces cerevisiae* and *Schizosaccharomyces pombe*). Using the data obtained by our analysis, it was possible to predict the MO required in a situation of replication stress. Also, our findings allow establishing a threshold for the number of origins, which serves as a parameter for genome approaches that map origins. Moreover, our data suggest that when compared to yeasts, trypanosomatids use much more origins than the minimum needed. This is the first time a comparative analysis of the minimum number of origins has been successfully applied. These data may provide new insight into the understanding of the replication mechanism and a new methodological framework for studying single-celled eukaryote genomes.

## 1. Introduction

In cellular organisms, DNA replication is a crucial and highly regulated process that follows specific steps, which vary slightly between prokaryotes and eukaryotes. In general, the earliest step in DNA replication is the establishment of replication origins, i.e., the genomic loci where DNA synthesis begins [1]. The start of replication is preceded by the binding of an initiator at the replication origins, which recruits and activates the replisome in a process called origin firing. Each origin fired produces two replication forks in opposite directions (bidirectional movement), which are responsible for synthesizing DNA at a rate that varies according to the organism and cell type [1,2,3]. The replication time required for all chromosomes determines the S-phase duration. Although the S-phase length is referred to as a way of regulating the cell cycle progression [4,5], recent studies have suggested that it is extremely robust [6,7,8].

Studies indicated that bacteria [1,9] and some protozoan parasites, such as *Leishmania* spp. [10], typically have one single origin per chromosome. On the other hand, most other eukaryotes, such as *S. cerevisiae* and *S. pombe*, generally have multiple origins per chromosome [9,11,12,13]. The exact number of origins per chromosome can vary according to cell type and the cellular environment [14]. However, the minimum number of origins (MO) required to duplicate an entire chromosome within a specific S-phase duration must show minimal variation because it depends on two very constant factors: average replication rate and chromosome size [8].

In trypanosomatids, single-celled eukaryotes that encompass human pathogens are of great medical importance, and the question about how many origins are needed to replicate an entire chromosome during the S-phase is totally open [8,10,15,16]. Even for the widely studied domain Bacteria and the model eukaryote *S. cerevisiae*, this discussion is not yet a closed subject [1,17,18,19].

Here, we developed a website that is able to determine the duration of each cell cycle phase—G1, S, G2, mitosis (M), and cytokinesis (C)—in any organism. After using this website to obtain the S-phase duration for the organisms analyzed, we applied a formula developed by our group [8] and showed a comparative analysis between the minimum number of origins (MO) in trypanosomatids (*T. cruzi*, *L. major*, and *T. brucei*) and yeasts (*S. cerevisiae* and *S. pombe*). In addition to contributing to a discussion of why some organisms use far more origins than the minimum required, this study provides a clue about the dynamic of replication during the S-phase, raising questions about the possible phenomena involved in this process.

## 2. Materials and Methods

### 2.1. Trypanosomatids Culture, Growth Curves, and Morphological Patterns

Epimastigote forms of *T. cruzi* (CL Brener strain) were cultured at 28 °C in liver infusion tryptose (LIT) medium supplemented with 10% (*v*/*v*) fetal bovine serum and 1% (*v*/*v*) antibiotic/antimycotic solution. Promastigote forms of *L. major* (strain Friedlin) were cultured at 26 °C in an M199 medium supplemented with 10% (*v*/*v*) heat-inactivated fetal calf serum, 25 mM HEPES, and 1% (*v*/*v*) antibiotic/antimycotic solution.

For the growth curves, each parasite culture was initiated with 1 × 10^6^ cells.mL^−1^. Each growth curve was harvested and counted daily until it reached the stationary phase. For the establishment of the morphological patterns, formaldehyde-fixed and DAPI-stained exponentially growing parasites (*T. cruzi* and *L. major*) were examined under an Olympus BX51 fluorescent microscope (Olympus, Tokyo, Japan) (100× oil objective) to observe the profile of organelles that contain DNA (nucleus and kinetoplast).

### 2.2. EdU Incorporation Assays and ‘Click’ Chemistry Reaction

Exponentially growing parasites were incubated with 100 µM 5-ethynyl-2′-deoxyuridine (EdU) (ThermoFisher Scientific, Waltham, MA, USA) for the time required according to each assay at species-specific temperatures (28 °C for *T. cruzi* and 26 °C for *L. major*). The parasites were then harvested by centrifugation at 2500 *g* for 5 min, washed three times in 1× PBS (137 mM NaCl, 2.7 mM KCl, 10 mM Na_2_HPO_4_, and 2 mM KH_2_PO_4_, pH 7.4), and the pellet was resuspended in 200 µL of the same buffer solution. Afterward, 100 µL of the cell suspension was loaded onto poly-L-lysine-pretreated microscope slides (Tekdon, Myakka, FL, USA), fixed for 20 min using 4% sterile paraformaldehyde (Merck, Darmstadt, Germany) diluted in 1× PBS, washed three times with 1× PBS, and then washed three times with 3% BSA (Sigma-Aldrich, Saint Louis, MO, USA) diluted in 1× PBS. Then, parasites were permeabilized for 10 min with 0.1% sterile Triton X-100 (Sigma Aldrich, Saint Louis, MO, USA) diluted in 1× PBS, washed three times with 1× PBS, and then washed three times with 3% BSA in 1× PBS. To detect incorporated EdU, we used the Click-iT EdU detection solution for 45 min protected from light. The Click-iT EdU detection mix solution consisted of 25 µL 500 mM ascorbic acid (C_6_H_8_O_6_), 5 µL 100 mM copper sulfate (CuSO_4_), 2.5 µL Alexa fluor azide 488 (ThermoFisher Scientific, Waltham, MA, USA), and 467.5 µL distilled water (for details about EdU procedure, see ref. [20]). Finally, the parasites were washed five times with 1× PBS. Vectashield Mounting Medium (Vector, Burlingame, CA, USA) containing 4′,6-diamidino-2-phenylindole dihydrochloride (DAPI) was used as an antifade mounting solution and to stain nuclear and kinetoplast DNA. Images were acquired using an Olympus Bx51 fluorescent microscope (100× oil objective) attached to an EXFO Xcite series 120Q lamp and a digital Olympus XM10 camera with camera controller software Olympus Cell F (Olympus, Tokyo, Japan). Images were further analyzed using ImageJ software (National Institutes of Health, USA) to count the numbers of EdU-positive parasites, and the percentage of proliferating parasites was calculated for each sample relative to the total number of DAPI-positive parasites.

### 2.3. Development of the CeCyD Website and Analysis of the Cell Cycle

The website CeCyD (Cell Cycle Duration estimator) was developed using the Python programming language plus the Django v.1.8 framework. CeCyD is available at the following address http://cecyd.vital.butantan.gov.br/, and its source code is released under the GNU GPL-3 license at https://github.com/bruno-fs/CeCyD.

To estimate the duration of mitosis (M) and cytokinesis (C), the CeCyD uses the Williams (1971) equation [21]:
(1)$$x=\frac{\ln(1-y/2)}{-\alpha }$$
where *x* is the cumulative time within the cell cycle necessary to reach the start of the phase in question, i.e., the difference between the doubling time and *x* will give the time of the remaining phase(s); *y* is the cumulative proportion of cells up the phase in question (expressed as a fraction of one unit), i.e., the difference between the total cells (1% or 100%) and the percentage of cells in C or M+C, will provide the *y* value for C and M, respectively. Finally, α is the specific growth rate.

To estimate the G2 phase, the CeCyD must receive from the user the value corresponding to the period required for a cell to pass through G2 and M phases. For this, the user must apply an EdU pulse (e.g., 15 min) and then collected parasites every 15 min until a single cell containing two EdU-labeled nuclei (2N2K in case of trypanosomatids) is observed. The difference between this value and the duration of mitosis previously calculated corresponds to the G2-phase duration.

The S-phase duration is estimated by the CeCyD according to the Stanners and Till (1960) equation [22]:
(2)$$S=\frac{1}{\alpha }\ln[L+e^{\alpha (Z)}]-(Z+t)$$
where *L* is the proportion of cells exhibiting EdU-labeled nuclei, α = ln 2/T (T = doubling time expressed in hours), *Z* = *G*2 + *M* + *C*, and *t* is the duration of the EdU labeling period in hours. Finally, the G1-phase duration is estimated by the difference between the doubling time and the sum of the remaining phases.

### 2.4. Estimation of the Minimum Number of Replication Origins (MO)

To estimate the MO needed to replicate an entire chromosome within the S-phase duration, we developed a mathematical inequation [8]. This formula uses as argument the S-phase duration (S) (which can be estimated by the CeCyD website), the size of the chromosome in question (N), and the replication rate (v). The lower bound MO for the number of origins required to replicate an entire chromosome is given by:
(3)$$mo\geq \lceil \frac{N}{2,v,S}\rceil ,$$

Of note, if the right-hand side of this inequation results in a fraction of a unit, then the next higher integer unit must be taken as the result of the inequation, which is represented by the ceiling function (⌈ ⌉).

For each organism analyzed, we used as parameters for the formula up-to-date data available in the TriTrypDB database (www.tritrypdb.org), NCBI database (www.ncbi.nlm.nih.gov), and data reported in the studies related [3,8,11,12,13,23,24,25,26] (see Table 1 for more details).

### 2.5. Origins Estimated by DNA Combing

To estimate how many origins are activated (on average) during the S-phase in any organism, we develop a simple mathematical equation. This equation uses a ratio between the size of the chromosome in question (N), and the inter-origin distance (IOD) obtained by DNA combing to estimate, on average, the total number of origins fired during the S-phase. The equation is given by:
(4)$$Oc=\lceil \frac{N}{IOD}\rceil ,$$

If the right-hand side of this equation results in a fraction of a unit, then the next higher integer unit must be taken as the result of the inequation, which is represented by the ceiling function (⌈ ⌉).

## 3. Results and Discussion

### 3.1. The CeCyD Website Allows a Quick Estimation of the Cell Cycle Phases Duration

Many studies have been using the two formulas developed by Williams (1971) [21] and Stanners and Till (1960) [22] to estimate the length of the cell cycle phases [8,27,28,29,30,31,32]. However, these estimations demand time and attention due to a large number of calculations involved. Also, they are subject to errors during the calculations. To facilitate the calculations and optimize the time consumed of these estimations, we developed a website called CeCyD (Cell Cycle Duration estimator), as shown in Figure 1A. CeCyD is available at the address http://cecyd.vital.butantan.gov.br/.

Briefly, CeCyD is a user-friendly website able to calculate the values of cytokinesis (C), mitosis (M), G2, S, and G1 phases of the cell cycle, for any organism. For this, the user needs the following experimental parameters: doubling time, percentage of cells in cytokinesis, percentage of cells in mitosis, minimum time to detect two EdU-labeled nuclei in the same cell, percentage of cells EdU-labeled after EdU pulse, and the duration of this EdU pulse.

To test and evaluate the efficiency of the CeCyD, we first obtained the parameters required for *L. major* and *T. cruzi* (CL Brener strain) from experimental analyses, as displayed in Appendix A. Then, we withdrew the same parameters for *T. brucei* from our previous study [8]. Next, we imputed the parameters in the CeCyD and estimated the duration of the cell cycle phases for each of these organisms, as shown in Figure 1B. *T. cruzi* presented G1 = 5.91 h (0.246 ccu), S = 9.86 h (0.411 ccu), and G2 + M + C = 8.23 h (0.343 cuu); *L. major* presented G1 = 5.52 h (0.53 ccu), S = 3.2 h (0.31 ccu), and G2 + M + C = 1.78 h (0.16 ccu); and *T. brucei* presented G1 = 3.37 h (0.397 ccu), S = 2.31 h (0.272 ccu), and G2 + M + C = 2.82 h (0.331 ccu). Of note, ccu means cell cycle unit, where one unit corresponds to the doubling time of each organism.

As expected, for *T. brucei*, the values provided by CeCyD were the same as those obtained in our previous work [8]. For both *L. major* and *T. cruzi*, when we compare the values provided by CeCyD with those obtained from other studies [30,32,33], we can observe similarities among the length of the cell cycle phases when EdU is used to monitor DNA replication [30]. However, when 5-bromo-2’-deoxyuridine (BrdU) is used to monitor DNA replication instead of EdU, the values obtained shown pronounced differences [33]. As already reported by our group [30], this discrepancy can be explained by the fact that there are differences in the detection of BrdU/EdU incorporation assays, i.e., EdU is more sensitive in monitoring DNA replication than the halogenated thymidine analogs (e.g., BrdU).

For our analyses, we used the S-phase duration from *T. cruzi*, *L. major*, *T. brucei*, *S. cerevisiae*, and *S. pombe*. Of note, for *T. brucei*, *S. cerevisiae,* and *S. pombe*, we did not use the CeCyD because the cell cycle phases duration for these organisms were already available [8,25,26,34,35], as shown in Figure 1B. Also, the cell cycle parameters used here were obtained from epimastigote cells of *T. cruzi*, promastigote cells of *L. major*, procyclic cells of *T. brucei*, mother cells of *S. cerevisiae,* and mitotic cells of *S. pombe*.

### 3.2. The Parameters Chromosome Size, S-Phase Duration, and Replication Rate Allow Estimating the MO per Chromosome in Any Organism

In a recent study, our group developed a formula able to estimate the MO required to duplicate an entire chromosome within the S-phase duration [8]. The development of this formula was based on the bidirectional movement of the replication forks, replication rate, S-phase duration, and the chromosome size in question. Although used only in *T. brucei* so far, this formula can be applied in any cell type. To demonstrate this, we estimated the MO in *T. cruzi*, *L. major*, *T. brucei* (using updated parameters), *S. cerevisiae*, and *S. pombe*, as shown in Table 1.

Among the single-celled eukaryotes analyzed here, *T. cruzi* draws attention because it is the only organism that requires only one origin per chromosome (MO = 1) to replicate its nuclear genome within the S-phase duration, as displayed in Table 1. *L. major*, on the other hand, requires more than one origin per chromosome to replicate its larger chromosomes (>1000 kb), while *T. brucei*, *S. cerevisiae*, and *S. pombe* requires more than one origin per chromosome to replicate their nuclear genomes, even for small chromosomes (<1000 kb), as shown in Table 1. As the formula to estimate the MO (Equation (3)) depends on the chromosome size, S-phase duration, and replication rate, the explanation for these organisms possess different MOs is related to these variables. For instance, *T. cruzi* has a long S-phase duration (9.86 h or 0.411 ccu) relative to other organisms analyzed here shown in Figure 1B, which justifies its MO per chromosome equaling 1, as presented in Table 1. Figure 1B and Table 1 show that *T. brucei* has an S-phase duration and replication rate similar to *L. major*; however, its chromosomes are larger than 1,000,000 bp (called megabase chromosomes [36]), which justifies the use of more than one origin per chromosome. *S. cerevisiae*, on the other hand, has a short S-phase duration (0.52 h or 0.347 ccu) and a low replication rate (1.6 kb·min^−1^ [11]), which imply, according to our formula, high MO values, as shown in Figure 1B and Table 1. *S. pombe*, in turn, has longer chromosomes, a short S-phase duration (0.4 h or 0.1 ccu) and a low replication rate (0.91 kb·min^−1^ [24]), which also imply high MO values. These data are also shown in Figure 1B and Table 1.

It is difficult to establish a reason why some organisms need a different number of origins during the S-phase. However, we can speculate that the number of origins needed to replicate all chromosomes during the S-phase is closely related to the S-phase duration itself. The question that remains is as follows: does the number of fired origins determine the S-phase duration, or is the S-phase duration robust, and a different number of origins is required to maintain this robustness? Although some studies point to robustness in S-phase duration [6,7,8], further studies are necessary to confirm which of these questions is the correct one.

It is worth to mention that among the parameters used to determine the MO, the replication rate is the most prone to alterations. Many factors can change the replication rate, such as decreased nucleotide pool [37,38], replication-transcription conflicts [39,40], DNA damage [41,42], among others, all of which leads mostly to some replication stress [43]. In other words, replication stress can be defined, in general, as the slowing of replication rate [44]. Thus, cells under replication stress probably would show different MO values relative to those estimated using the average replication rate from a wild type population.

### 3.3. In the Presence of Hypothetical Replication Stress, the MO Increase to Maintain Robustness in S-Phase Duration

To predict the behavior of MO in the presence of hypothetical replication stress, we simulated two conditions considering that the S-phase duration is robust [6,7,8]. The first one was mild replication stress, which was characterized here by a replication rate at 2/3 of the average value from the wild type population. The second situation was harsh replication stress, with a replication rate at a 1/3 of the average value. After applying these hypothetical values in Equation (3), we estimated the MO for the first (MO^MR^) and second (MO^HR^) conditions in trypanosomatids and *S. cerevisiae*, as shown in Table 2. It is worth mentioning that we did not perform this prediction for *S. pombe* because the peculiar behavior of its cell cycle seems to contribute to a flexible (non-robust) S phase duration [25,26].

According to our prediction, the only way a cell can maintain certain robustness in the S-phase duration in the presence of mild or harsh replication stress is to increase origin activation, which was evidenced by the increase in the MO values shown in Table 2. In other words, the demand for a higher number of activated origins in the presence of replication stress characterized by a slowing of replication rate can suggest that the cell tries to maintain robustness over the S-phase duration. This predicted behavior has already been evidenced by several cell types [45,46,47,48], including trypanosomatids [8] and *S. cerevisiae* [49]. However, *S. pombe* is an exception to this, because in addition to its S-phase not being robust [25,26], in the presence of replication stress, the origin firing is inhibited [24,50]. Moreover, *S. pombe* has other features that make its cell cycle unique when compared to other organisms: the S-phase is initiated before completion of the cytokinesis of the ongoing cell cycle [26,51], the cell mass influences the duration of the S-phase [25], and the main cell cycle control point is a size control in G2 phase [51]. Altogether, these peculiarities seem to contribute to more flexibility in the S-phase duration of this yeast. Nevertheless, further studies are still needed to better understand the dynamics of the replication stress response and origin usage through the S-phase in these non-metazoan organisms.

### 3.4. The MO Allows the Establishment of a Threshold That Can Serve as a Parameter by Other Methods That Detect Origins

To compare the MO with the origins obtained by different experimental approaches, we set up graphs in order to show trend lines for each methodology analyzed, as shown in Figure 2A–E. We observed an expected positive correlation between the number of origins and the size of each chromosome, i.e., the larger the chromosome, the more origins are required to replicate it within the S-phase duration. As the MO is estimated from relatively constant parameters in a wild type population, the trend line of the MO, shown in Figure 2A–E in black lines, allows the establishment of a threshold that can serve as a parameter when estimating the number of origins by other methods.

Before we go on with our analysis, it is worth mentioning that according to their different usages, replication origins can be classified into three categories: constitutive, which are always activated in all cells of a given population; flexible, whose usage varies from cell to cell; or dormant, which are not fired during a normal cell cycle but are activated in the presence of replication stress [52]. However, due to the technical difficulty in distinguishing flexible and dormant origins, we refer to these only as non-constitutive origins.

Thus, when comparing the trend line of the origins estimated by DNA combing (the red lines in Figure 2A–E) with the trend lines of MO (the black lines in Figure 2A–E,), we observed that *T. cruzi*, *L. major*, *T. brucei*, *S. cerevisiae*, and *S. pombe* use, on average, more origins per chromosome than the minimum required, i.e., the red lines are above from the black ones, as shown in Figure 2A–E). This makes sense, given that the DNA combing approach estimates the pool of all origins (constitutive + non-constitutive) fired in a population.

For *L. major*, in addition to the trend line for origins estimated by DNA combing (the red line in Figure 2B), we also plotted a trend line for potential origins mapped by small leading nascent strand purification coupled to next-generation sequencing (SNS-seq) [16] (the blue line in Figure 2B), and a trend line for origins mapped by marker frequency analysis (MFA-seq) [10] (the green line in Figure 2B). The trend line of the potential origins mapped by SNS-seq is far above from the others (the blue line in comparison with the others in Figure 2B). This makes sense because the SNS-seq approach has high accuracy and resolution in detecting small sites of replication, which can include DNA repair, potential origins, and other events that generate DNA synthesis in a population. However, the trend line of origins mapped by MFA-seq is below the threshold imposed by the minimal origins (MO trend line) (the black line in comparison with the green one in Figure 2B). This implies that only with origins mapped by MFA-seq [10], *L. major* is not able to replicate its nuclear genome within the S-phase duration. Although it seems meaningless, this can be easily explained by the fact that the MFA-seq analysis has low resolution and accuracy, probably being able to identify only the constitutive origins in a population and not the entire pool of fired origins, as occurs in the DNA combing approach for example [3,8]. Nevertheless, further studies are necessary to figure out how many origins are indeed used for a single cell of *L. major* during a standard cell cycle.

In *T. brucei*, we also plotted a trend line for the origins mapped by MFA-seq, and the situation is similar to that presented by *L. major* (the black dots in comparison with the green ones in Figure 2C), i.e., some chromosomes are not able to be duplicated only with the origins mapped by MFA-seq, as already explained in a recent study of our group [8]. Briefly, the reason is the same as previously explained: the low resolution of the MFA-seq analysis.

For *S. cerevisiae*, in addition to plotting a trend line for origins estimated by DNA combing [53] (the red line in Figure 2D) and a trend line for origins mapped by microarray analysis [12] (the green line in Figure 2D), we also plotted a trend line for the known positions of origins [54] (the purple line in Figure 2D). The known position of origins refers to a conserved DNA sequence (called autonomous replicating sequences—ARS) where the assembly of the pre-replication complex occurs [53,55,56]. The trend line of the known origins is above from the MO trend line (the purple line in comparison with the black one in Figure 2D), and above from the trend line of the origins estimated by DNA combing (the purple line in comparison with the red one in Figure 2D). This makes sense because the known positions of origins are potential sites for the establishment of origins. However, not all of these sites are activated during the S-phase, i.e., in *S. cerevisiae*, there are many more potential sites for the establishment of origins than those that are indeed used to complete replication within the S-phase duration [19,57,58,59,60]. The trend line of origins mapped by microarray analysis is above the threshold imposed by the minimum origins (MO trend line) (the green line in comparison with the black one in Figure 2D), but below the trend lines of both the known origins and the origins estimated by DNA combing (the comparison amongst the green, purple and red lines in Figure 2D). This was expected for the same reason raised before, i.e., just like MFA-seq, microarray analysis also has low accuracy in detecting the entire pool of origins activated in a population and maps mainly the constitutive origins.

Unlike *S. cerevisiae*, *S. pombe* lacks a consensus DNA sequence that determines origin sites. However, its origins coincide with chromosomal A+T-rich islands [13]. Thus, in addition to plotting a trend line for origins estimated by DNA combing [50] (the red line in Figure 2E) and a trend line for origins mapped by microarray analysis [61] (the yellow line in Figure 2E), we also plotted a trend line for the A+T rich islands [13] (the gray line in Figure 2E). All three of these trend lines (red, yellow, and gray) are above from the threshold imposed by the minimum origins (MO trend line) and practically overlap each other, although as the comparison amongst all the trend lines in Figure 2E shows, the trend line of the origins estimated by DNA combing is slightly above, as expected. This overlapping of the trend lines raises a question about the dynamics of origin usage during the S-phase in *S. pombe*, which seems to be relatively peculiar when compared to other single-celled eukaryotes [50].

Of note, so far, there is no data about MFA-seq or microarray analysis in *T. cruzi*, which prevents a deeper comparative analysis in this organism.

### 3.5. Trypanosomatids Can Use Around Fivefold More Origins than the Minimum Required to Complete Replication within the S-Phase Duration

To investigate how many times more origins than the minimum (MO) the organisms analyzed can use, we calculated the ratio between the angular coefficient (a value) of linear equations (y = ax + b) of maximum origins used and MO, shown as the red and black lines, respectively, in Figure 2. Here, we defined maximum origins used as the origins estimated by DNA combing, represented by the red lines in Figure 2; see material and methods.

Using this reasoning, we can estimate that *S. pombe* uses, on average, 1.44 times more origins than the MO, while in *S. cerevisiae*, this ratio is 2.12. Interestingly, in trypanosomatids, this ratio is higher. In *T. brucei* this ratio is 4.91, in *L. major* is 5.1, and in *T. cruzi* this ratio is so high that tends to infinity since *T. cruzi* needs only one origin per chromosome to replicate its nuclear genome (MO = 1 for all chromosomes), i.e., the MO linear equation is y = 1, as shown in Figure 3.

Although we cannot classify the total origins used as constitutive or non-constitutive, one question can be raised: what makes trypanosomatids apparently use a pool of origins much higher than the MO when compared to the yeasts *S. cerevisiae* and *S. pombe*? One possible explanation is that in trypanosomatids, unlike other eukaryotes, the majority of their genes are organized into large polycistronic clusters, which could favor replication stress through replication–transcription conflicts [8]. Replication stress, as reported in some studies [48,62], is a potential contributor for the activation of replication origins. However, although proposed by our group [8], this hypothesis needs to support more experimental assays to gain credibility. Another possibility is that the replication rate of *S. cerevisiae* and *S. pombe* are lower than those in trypanosomatids (1.6 kb·min^−1^ in *S. cerevisiae*, 0.91 kb·min^−1^ in *S. pombe*, and 2–3 kb·min^−1^ in trypanosomatids), as shown in Table 1. *S. cerevisiae* has a chromosomes size and an S-phase duration similar to those found in trypanosomatids shown in Table 1 and Figure 1B. Thus, the only way to maintain robustness in the S-phase duration is by activating more origins. Apparently, *S. cerevisiae* does just that, but further studies are necessary to figure out its exact dynamics of origin usage during the S-phase. On the other hand, *S. pombe* has larger chromosomes and a relatively short S-phase duration when compared to trypanosomatids, as displayed in Table 1 and Figure 1B. Moreover, as already mentioned, *S. pombe* does not have a robust S-phase [25,26] and its origins fire stochastically [50], which precludes any speculation regarding its peculiar dynamics of origin usage. However, unlike trypanosomatids, *S. pombe* appears to use a number of origins very close to the minimum required.

This is the first time a comparative analysis of the minimum number of origins has been successfully applied. These data may provide new insight into the understanding of origin usage during the S-phase and a new methodological framework for studying single-celled eukaryotes genomes.

## 4. Conclusions

Here, we demonstrate that the minimum number of origins (MO) required to duplicate an entire chromosome within the S-phase duration can be easily obtained from the parameters chromosome size, S-phase duration, and replication rate. Predictions performed by us suggest that in the presence of replication stress, all the organisms analyzed here demands higher MO values. Moreover, we evidenced here that the MO allows the establishment of a threshold that can serve as a parameter by other methods that detect origins. Also, our data strongly suggest that trypanosomatids can use around fivefold more origins than the MO. This value is relatively higher than other single-celled organisms, such as the yeasts *S. cerevisiae* and *S. pombe*. However, further studies are required to figure out the dynamics of origin usage during the S-phase in these organisms, especially in trypanosomatids.

## Figures and Tables

**Figure 1 genes-11-00523-f001:**
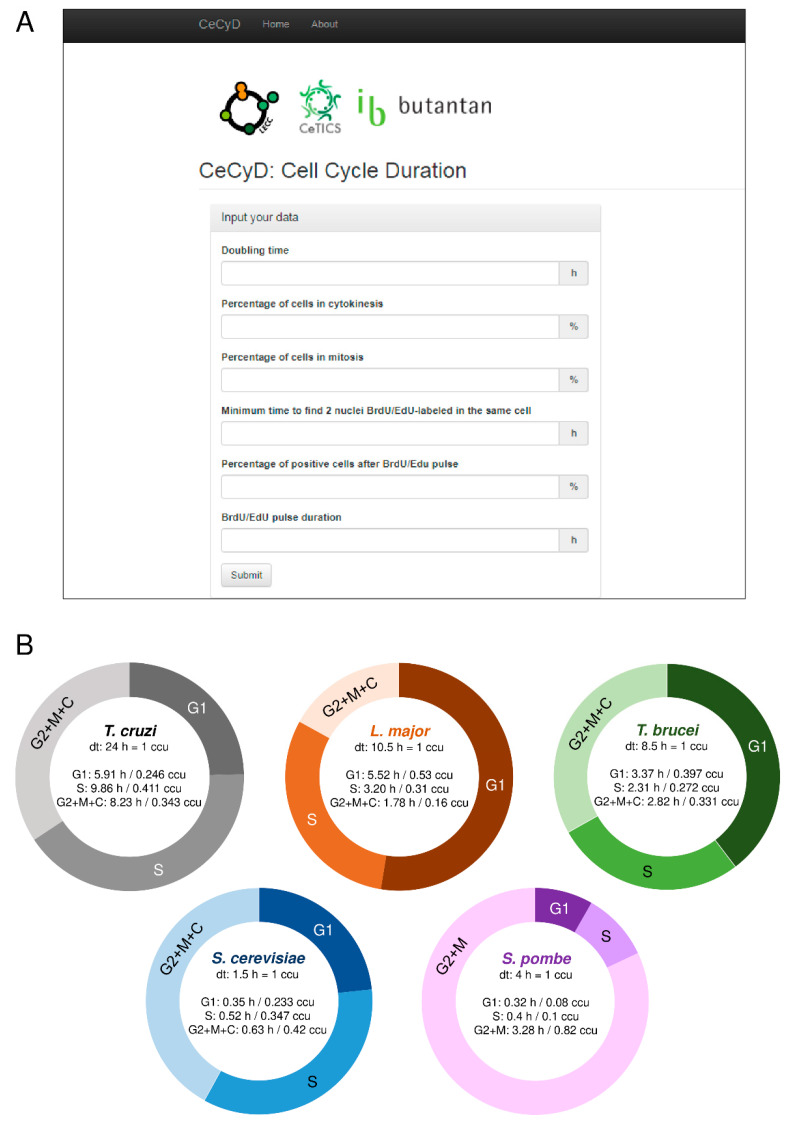
The user-friendly website CeCyD allows a quick estimation of the cell cycle phases duration for any cell type. (**A**) Screenshot of the CeCyD website showing the parameters to be inserted. This website is available at the address http://cecyd.vital.butantan.gov.br/. (**B**) Estimation of the cell cycle phases lengths (G1, S, and G2+M+C/G2+M) for *T. cruzi*, *L. major*, *T. brucei*, *S. cerevisiae*, and *S. pombe*. For *T. cruzi*, *L. major*, and *T. brucei* from calculations made using the CeCyD website. For *S. cerevisiae* and *S. pombe*, the values were obtained from other studies [25,26,34,35].

**Figure 2 genes-11-00523-f002:**
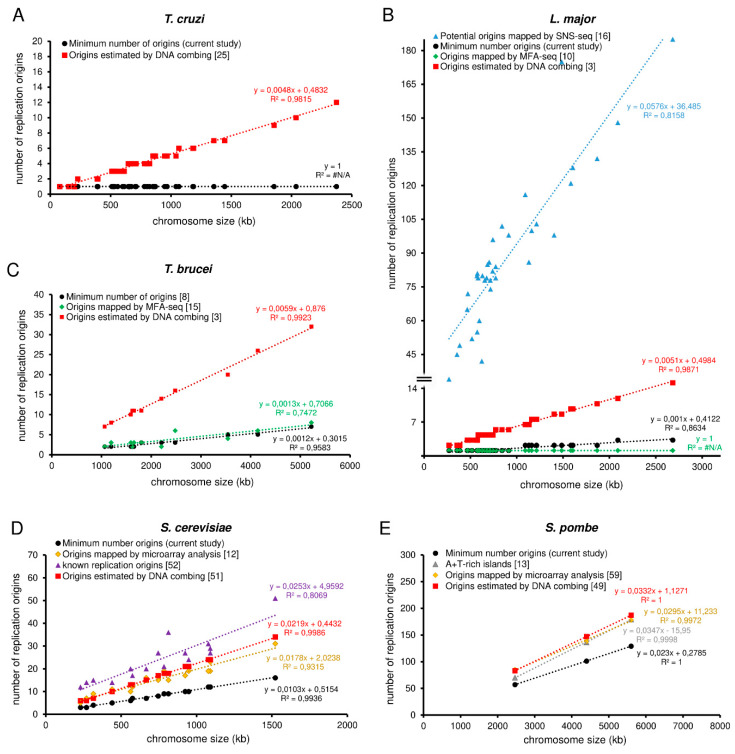
Comparative analysis among different approaches used to estimate replication origins in trypanosomatids (*T. cruzi*, *L. major*, *T. brucei*) and yeasts (*S. cerevisiae* and *S. pombe*) (a–e). Graphs showing positive correlations between chromosome length and the number of replication origins estimated by different approaches: minimum number of origins—MO (black), origins estimated by DNA combing (red), origins estimated by MFA-seq (green), potential origins mapped by SNS-seq (blue), origins estimated by microarray (yellow), known origins (purple), and A+T rich islands (gray). (**A**) *T. cruzi*, (**B**) *L. major*, (**C**) *T. brucei*, (**D**) *S. cerevisiae*, and (**E**) *S. pombe*. The trend lines for all approaches, as well as the equations, are shown. Studies are referenced in each graph.

**Figure 3 genes-11-00523-f003:**
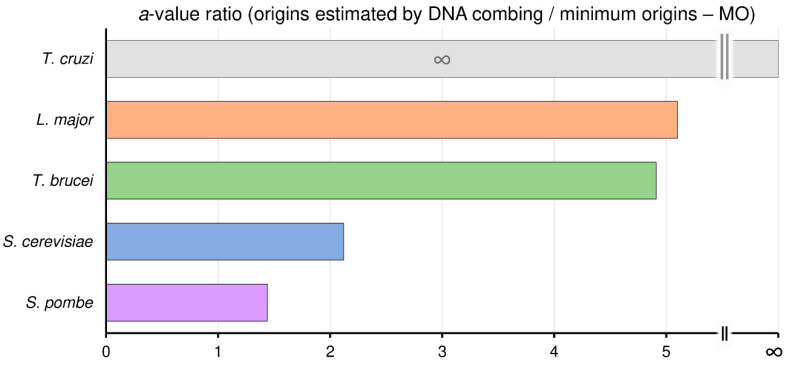
Trypanosomatids use around fivefold more origins than the minimum required. Angular coefficient (a-value) ratios between origins estimated by DNA combing and the minimum origins (MO) for *T. cruzi* (gray bar), *L. major* (orange bar), *T. brucei* (green bar), *S. cerevisiae* (blue bar), and *S. pombe* (purple bar).

**Table 1 genes-11-00523-t001:** Calculation of the minimum number of origins (MO) per chromosome in trypanosomatids (*T. cruzi*, *L. major*, and *T. brucei*) and yeasts (*S. cerevisiae* and *S. pombe*).

Chrom.	*T. cruzi* ^1^		*L. major* ^2^		*T. brucei* ^3^		*S. cerevisiae* ^4^		*S. pombe* ^5^	
	Size (bp)	MO	Size (bp)	MO	Size (bp)	MO	Size (bp)	MO	Size (bp)	MO
I	77,958	1	268,988	1	1,064,672	2	230,19	3	5,598,923	129
II	151,740	1	355,712	1	1,193,948	2	813,14	9	4,397,795	101
III	196,644	1	384,502	1	1,653,225	2	315,34	4	2,465,919	57
IV	200,401	1	472,852	1	1,590,432	2	1,522,19	16	-	-
V	227,319	1	465,823	1	1,802,303	2	574,86	7	-	-
VI	389,024	1	516,869	1	1,618,915	2	270,15	3	-	-
VII	391,095	1	596,352	1	2,205,233	3	1,090,94	12	-	-
VIII	393,423	1	574,960	1	2,481,190	3	562,64	6	-	-
IX	509,634	1	573,434	1	3,542,885	4	439,88	5	-	-
X	518,846	1	570,865	1	4,144,375	5	745,44	8	-	-
XI	526,141	1	582,573	1	5,223,313	6	666,45	7	-	-
XII	533,093	1	675,346	1	-	-	1,078,17	12	-	-
XIII	558,364	1	654,595	1	-	-	924,43	10	-	-
XIV	598,625	1	622,644	1	-	-	784,33	9	-	-
XV	612,853	1	629,517	1	-	-	1,091,28	12	-	-
XVI	646,207	1	714,651	1	-	-	948,06	10	-	-
XVII	648,584	1	684,829	1	-	-	-	-	-	-
XVIII	655,081	1	739,748	1	-	-	-	-	-	-
XIX	671,453	1	702,208	1	-	-	-	-	-	-
XX	656,799	1	742,537	1	-	-	-	-	-	-
XXI	704,149	1	772,972	1	-	-	-	-	-	-
XXII	710,778	1	716,602	1	-	-	-	-	-	-
XXIII	655,477	1	772,565	1	-	-	-	-	-	-
XXIV	779,922	1	840,950	1	-	-	-	-	-	-
XXV	822,374	2	912,845	1	-	-	-	-	-	-
XXVI	801,422	1	1,091,540	2	-	-	-	-	-	-
XXVII	850,241	2	1,130,424	2	-	-	-	-	-	-
XXVIII	853,233	1	1,160,104	2	-	-	-		-	-
XXIX	870,934	1	1,212,663	2	-	-	-	-	-	-
XXX	863,882	1	1,403,434	2	-	-	-	-	-	-
XXXI	947,473	1	1,484,328	2	-	-	-	-	-	-
XXXII	968,069	1	1,604,637	2	-	-	-	-	-	-
XXXIII	1,041,172	1	1,583,653	2	-	-	-	-	-	-
XXXIV	1,065,764	1	1,866,748	2	-	-	-	-	-	-
XXXV	1,186,946	1	2,090,474	3	-	-	-	-	-	-
XXXVI	1,180,744	1	2,682,151	3	-	-	-	-	-	-
XXXVII	1,355,803	1	-	-	-	-	-	-	-	-
XXXVIII	1,444,805	1	-	-	-	-	-	-	-	-
XXXIX	1,854,104	1	-	-	-	-	-	-	-	-
XL	2,036,760	1	-	-	-	-	-	-	-	-
XLI	2,371,736	1	-	-	-	-	-	-	-	-

^1^*T. cruzi*: S-phase duration = 591.6 min (current study), replication rate = 2.05 kb·min^−1^ [23]; ^2^
*L. major*: S-phase duration = 192 min (current study), replication rate = 2.44 kb·min^−1^ [3]; ^3^
*T. brucei*: S-phase duration = 138.6 min [8], replication rate = 3.06 kb·min^−1^ [8]; ^4^
*S. cerevisiae*: S-phase duration = 30 min [34,35], replication rate = 1.6 kb·min^−1^ [11]; ^5^
*S. pombe*: mitotic S-phase duration = 24 min [25,26], mitotic replication rate = 0.91 kb·min^−1^ [24].

**Table 2 genes-11-00523-t002:** Calculation of the minimum number of origins (MO) per chromosome in the presence of mild (MO^MR^) and harsh (MO^HR^) replication stress.

Chrom.	*T. cruzi* ^1^			*L. major* ^2^			*T. brucei* ^3^			*S. cerevisiae* ^4^		
	MO	MO^MR^	MO^HR^	MO	MO^MR^	MO^HR^	MO	MO^MR^	MO^HR^	MO	MO^MR^	MO^HR^
I	1	1	1	1	1	1	2	2	4	3	4	8
II	1	1	1	1	1	2	2	3	5	9	13	26
III	1	1	1	1	1	2	2	3	6	4	5	10
IV	1	1	1	1	1	2	2	3	6	16	24	48
V	1	1	1	1	1	2	2	4	7	7	10	19
VI	1	1	1	1	1	2	2	3	6	3	5	9
VII	1	1	1	1	1	2	3	4	8	12	18	35
VIII	1	1	1	1	1	2	3	5	9	6	9	18
IX	1	1	1	1	1	2	4	7	13	5	7	14
X	1	1	1	1	1	2	5	8	15	8	12	24
XI	1	1	1	1	1	2	6	10	19	7	11	21
XII	1	1	1	1	2	3	-	-	-	12	17	34
XIII	1	1	1	1	2	3	-	-	-	10	15	30
XIV	1	1	1	1	1	3	-	-	-	9	13	25
XV	1	1	1	1	2	3	-	-	-	12	18	35
XVI	1	1	1	1	2	3	-	-	-	10	15	30
XVII	1	1	1	1	2	3	-	-	-	-	-	-
XVIII	1	1	1	1	2	3	-	-	-	-	-	-
XIX	1	1	1	1	2	3	-	-	-	-	-	-
XX	1	1	1	1	2	3	-	-	-	-	-	-
XXI	1	1	1	1	2	3	-	-	-	-	-	-
XXII	1	1	1	1	2	3	-	-	-	-	-	-
XXIII	1	1	1	1	2	3	-	-	-	-	-	-
XXIV	1	1	1	1	2	3	-	-	-	-	-	-
XXV	1	1	2	1	2	3	-	-	-	-	-	-
XXVI	1	1	1	2	2	4	-	-	-	-	-	-
XXVII	1	1	2	2	2	4	-	-	-	-	-	-
XXVIII	1	1	2	2	2	4	-	-	-	-		-
XXIX	1	1	2	2	2	4	-	-	-	-	-	-
XXX	1	1	2	2	3	5	-	-	-	-	-	-
XXXI	1	1	2	2	3	5	-	-	-	-	-	-
XXXII	1	1	2	2	3	6	-	-	-	-	-	-
XXXIII	1	1	2	2	3	6	-	-	-	-	-	-
XXXIV	1	1	2	2	3	6	-	-	-	-	-	-
XXXV	1	1	2	3	4	7	-	-	-	-	-	-
XXXVI	1	1	2	3	5	9	-	-	-	-	-	-
XXXVII	1	1	2	-	-	-	-	-	-	-	-	-
XXXVIII	1	1	2	-	-	-	-	-	-	-	-	-
XXXIX	1	2	2	-	-	-	-	-	-	-	-	-
XL	1	2	2	-	-	-	-	-	-	-	-	-
XLI	1	2	2	-	-	-	-	-	-	-	-	-

^1^*T. cruzi*: S-phase duration = 591.6 min (current study), replication rate = 2.05 kb·min^−1^ [23]; ^2^
*L. major*: S-phase duration = 192 min (current study), replication rate = 2.44 kb·min^−1^ [3]; ^3^
*T. brucei*: S-phase duration = 138.6 min [8], replication rate = 3.06 kb·min^−1^ [8]; ^4^
*S. cerevisiae*: S-phase duration = 30 min [34,35], replication rate = 1.6 kb·min^−1^ [11].

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
