# Peer review of "Comparative Analysis of the Minimum Number of Replication Origins in Trypanosomatids and Yeasts"

_genes, 2020, doi:10.3390/genes11050523_

Round 1

Reviewer 1 Report

In this work, da Silva et al developed a web site to estimate the duration of the cell cycle phases of any unicellular organism from which only few experimental measurements are required. Then, they use the calculated S-phase length and the replication rates to infer the minimum number of replication origins necessary to accommodate full replication of two Trypanosomatid sp. and discuss their findings in comparison with Trypanosoma brucei and Saccharomyces cerevisiae.

The development of the web site is a useful tool to the scientific community as it provides an easy-to-use platform to perform cell cycle length comparisons between organisms and between cell populations grown in a variety of conditions. The estimation of the minimum number of origins required to complete replication in each organism is an extension from the previous work of the group, were they nicely showed that T. brucei need to activate more replication origins than previously established to complete S-phase. The methodology is well explained and the results are clear, well presented and discussed. I only have few comments to improve the work:

  1. When comparing the number of origins estimated by different methodologies in each of the organisms studied (Fig 2), the authors missed quoting the data from a previous work estimating more than one replication origin per chromosome (Lombraña et al., 2016, doi: 10.1016/j.celrep.2016.07.007). The authors should incorporate these data and discuss them along with their findings here.

  1. Similarly, it would be of interest to add to the study the other yeast model, Schizosaccharomyces pombe, from which origin positions, S-phase length and fork rates are well known (Nasmyth et al, 1979, PMID: 528581; Segurado et al., 2003, DOI:1038/sj.embor.embor7400008; Iyer and Rhind, 2017, doi: 10.1371/journal.pgen.1006958).

  1. Furthermore, although probably out of the scope of the current work, it will make their findings much more compelling if the authors try to test some predictions derived from their analysis. For example, can they infer the origin numbers required to replicate the brucei genome upon replicative stress conditions?

Reviewer 2 Report

The manuscript by da Silva et al. deals with DNA replication in the unicellular eukaryotes: trypanosomatids and budding yeast. Specifically, the manuscript introduces an online tool developed by the authors to calculate the length of cell cycle phases in all eukaryotes, as long as specific empirical data is supplied by the user. Then they use this tool to aid analysis of the minimum number of replication origins needed by trypanosomatids and budding yeast during S-phase and contrasting this data with epirical data obtained from these organisms when available.

Overall, the paper is interesting and the online tool developed by the authors has the potential to be used by researchers studying various unicellular organisms. The quality of the English language is initially good, but slowly deteriorates further into the manuscript. The authors should carefully control these later sections. Some language problems are pointed out below, but I think the text should also be controlled for language, particularly the ‘Results and Discussion’ and ‘Conclusions’.

Below are some points the authors should address:

In reading the paper, I was wondering why the authors did not try to text their software with data they had obtained in reference 8. It was only when I looked into that reference did I see they already did the type of analysis, which I assume motivated them to develop their online tool. This should be stated.  

Equations should be numbered.

Description (lines 101-103) of equation in line 100 is too generic. Additional text on how this equation works in context of examined trypanosomatids would clarify. Description of equation in line 111 (112-115) does this nicely.

Abbreviate “minimum number of replication origins” “MO”, not “mo”

Line 142: “…the time consumed by these…”

Line 144-145: source code info should go into Materials and Methods

Description of cell cycle unit should be moved from Fig 1 legend to main text when first mentioned (line 154)

Line 183: delete “Of note,”

Ref 29 used as a source of budding yeast data determined DA replication in a different way than the authors did in this study. Is it possible for the authors to find a reference in which DNA replication was dertemined with EdU istead of H3 uracil. If no, authors should mention this difference and why this may or may not make a difference in their calculations. Also, other sources are cited for budding yeast data in Table 1. Why?

Line 218: “…possessing different MOs…”

Line 225: delete “in”

Lines 228-9: “…is the S-phase duration robust…”

Run-on sentence in lines 254-6 awkward and hard to understand. Rewrite.

Line 259: “…the trend line of MOs…”; in fact, change all instances of “tendency line” to “trend line”

Figure 2 citations in graphs should use the same citation numbers as main text.

Line 278: delete “to”

Line 286: “genome-wide” what? Which method? Same for lines 292-3.

Regarding this statement Line 351: “…unlike eukaryotic model organisms…” . Trypansomatids are eukaryotic model organisms too! Restate this sentence.

Round 2

Reviewer 1 Report

The authors have answered all my requests in the revised version of their manuscript. The work has largely improved and their findings are more compelling for the readers.

I still want to make a couple of points:

  1. There is a mistake in the MO estimates for major in Table 1. It was not present in the previous version, so likely the mistake arose during the formatting process. Please check.
  2. Similarly, in the merged pdf I wasn´t able to see the bottom part of figures 2 and 3. I could follow the result in the text, though.
  3. In their interpretation of the distinct features of S. pombe replication, the authors should take into account that the unique cell cycle of this yeast is due to the fact that it is a “false diploid”, so DNA replication is initiated before completion of the cytokinesis of the ongoing cell cycle. The major cell cycle control is in G2/M, making this transition the rate-limiting step in their cell cycle and likely allowing more flexibility in the S-phase.

Some small points:

  • Line 43; MO
  • Line 164: other
  • Legend Fig 2: the newly incorporated SNS-seq data is not annotated in the legend
